# Research on LiDAR-Assisted Optimization Algorithm for Terrain-Aided Navigation of eVTOL

**DOI:** 10.3390/s25185672

**Published:** 2025-09-11

**Authors:** Guangming Zhang, Jing Zhou, Zhonghang Duan, Weiwei Zhao

**Affiliations:** School of Flight Technology, Civil Aviation Flight University of China, Guanghan 618307, China; gmzhang@cafuc.edu.cn (G.Z.); duanzhonghang@cafuc.edu.cn (Z.D.); zww@cafuc.edu.cn (W.Z.)

**Keywords:** electric vertical takeoff and landing aircraft, terrain-aided navigation, LiDAR, digital elevation model, multi-sensor fusion

## Abstract

To address the high-precision navigation requirements of urban low-altitude electric vertical take-off and landing (eVTOL) aircraft in environments where global navigation satellite systems (GNSSs) are denied and under complex urban terrain conditions, a terrain-matching optimization algorithm based on light detection and ranging (LiDAR) is proposed. Given the issues of GNSS signal susceptibility to occlusion and interference in urban low-altitude environments, as well as the error accumulation in inertial navigation systems (INSs), this algorithm leverages LiDAR point cloud data to assist in constructing a digital elevation model (DEM). A terrain-matching optimization algorithm is then designed, incorporating enhanced feature description for key regions and an adaptive random sample consensus (RANSAC)-based misalignment detection mechanism. This approach enables efficient and robust terrain feature matching and dynamic correction of INS positioning errors. The simulation results demonstrate that the proposed algorithm achieves a positioning accuracy better than 2 m in complex scenarios such as typical urban canyons, representing a significant improvement of 25.0% and 31.4% compared to the traditional SIFT-RANSAC and SURF-RANSAC methods, respectively. It also elevates the feature matching accuracy rate to 90.4%; meanwhile, at a 95% confidence level, the proposed method significantly increases the localization success rate to 96.8%, substantially enhancing the navigation and localization accuracy and robustness of eVTOLs in complex low-altitude environments.

## 1. Introduction

With the deep integration of unmanned aerial vehicle (UAV) technology and the new generation of information technology, the low-altitude economy has become increasingly prominent in promoting regional economic development, improving social and public services, and strengthening national defense security. As the core carrier of the low-altitude economy, electric vertical take-off and landing (eVTOL) aircraft, with their vertical take-off and landing capabilities and adaptability to urban environments, demonstrate significant application potential in scenarios such as logistics and transportation, emergency rescue, and geographic surveying and mapping. They are regarded as a key component of the future three-dimensional urban transportation system [1]. However, eVTOLs face severe challenges in low-altitude urban environments characterized by complex urban terrain and dense buildings that lead to intricate terrain features [2]. In densely built urban areas, global navigation satellite systems (GNSSs) are affected by signal obstruction and multipath effects, resulting in positioning errors that can reach tens of meters. Although inertial navigation systems (INSs) can provide short-term autonomous positioning, its errors accumulate over time, making it unable to meet the high-precision navigation requirements for long-range flights. In complex urban environments where GNSSs are denied, terrain-aided navigation (TAN) [3] technology offers advantages such as independence and immunity to adverse weather conditions, making it a key solution for providing absolute position corrections and suppressing the cumulative errors of INSs. TAN technology works by acquiring real-time elevation information of the terrain beneath the aircraft to generate a dynamic digital elevation model (DEM), which is then matched with a pre-stored reference DEM to correct the cumulative errors of the INS [3]. Among current terrain-matching methods, those based on ranging sensors and terrain features [4,5,6] show significant potential but still have limitations. For instance, the scale-invariant feature transform (SIFT) algorithm experiences a sharp decline in the number of extracted feature points in low-texture areas (such as flat road surfaces or regular building rooftops) [7], severely affecting positioning reliability. Meanwhile, the random sample consensus (RANSAC) algorithm, used for eliminating mismatched points, relies on the inlier rate and the setting of iteration numbers, leading to low efficiency or difficulty in finding the correct solution when the mismatch rate is high (low inlier rate), i.e., exhibiting poor matching performance in complex urban terrain scenarios.

The introduction of light detection and ranging (LiDAR) technology offers a breakthrough solution to the aforementioned issues [8,9]. LiDAR can directly acquire three-dimensional point cloud data with centimeter-level precision, enabling the construction of high-resolution real-time terrain elevation maps. This provides a reliable source of terrain information to enhance matching accuracy, circumventing the challenge of sparse features in low-texture areas (such as flat road surfaces and regular rooftops) encountered with optical images. Additionally, LiDAR technology does not rely on external lighting conditions and can operate stably in complex optical environments, including nighttime, low-light conditions, or smoke, making it suitable for a variety of complex environments with high navigation performance requirements. However, for real-time terrain-matched navigation applications in urban low-altitude eVTOL aircraft, high-resolution LiDAR generates massive amounts of point cloud data per second (ranging from hundreds of thousands to millions of points). The real-time filtering, feature extraction, feature description, and matching computations of this data impose enormous pressure on onboard computing resources, making it difficult to meet the high real-time requirements for navigation updates during eVTOL low-altitude flight. In urban environments, dense and highly dynamic elements (such as pedestrians, vehicles, and vegetation) introduce a significant amount of dynamic noise and non-ground points into the point cloud. Traditional feature extraction methods based on global features or fixed thresholds experience a marked decline in the effectiveness and robustness of feature extraction under dynamic noise interference, leading to an increased rate of mismatches in the matched point sets. This renders traditional random sample consensus (RANSAC) algorithms, which rely on a high inlier ratio, inefficient or even ineffective, thereby failing to ensure reliable position correction.

Research on LiDAR-based terrain-matching navigation for unmanned aerial vehicles (UAVs) in low-altitude environments has made significant progress in recent years. LiDAR technology can be applied in various aspects, including high-precision 3D terrain modeling [10,11,12], point cloud data processing and analysis [13,14,15,16], multi-sensor fusion navigation [17,18,19,20], terrain-aided navigation systems [21,22], and the comparison and optimization of [23,24]. Despite extensive research efforts aimed at enhancing the precision and efficiency of LiDAR terrain matching, in-depth investigation is still warranted regarding the core issue of efficient and robust feature extraction and matching from LiDAR point clouds in complex and dynamic urban low-altitude environments. This is particularly true in areas such as effectively suppressing dynamic noise interference, improving the usability of low-texture regions, and developing fast and reliable mismatch-removal algorithms under conditions of high mismatch rates.

At present, a substantial amount of research is centered on high-precision 3D modeling and point cloud processing algorithms. However, studies on the real-time and efficient processing of massive point clouds for eVTOL airborne platforms with limited onboard resources remain inadequate. Meanwhile, the robustness of existing methods against the prevalent dynamic noise sources and non-ground point interferences in urban environments has not been sufficiently addressed. For example, He et al. [25] focused primarily on the automatic registration problem of low-altitude LiDAR point clouds and aerial images, proposing an automatic registration method based on urban major road information. However, this method relies on the reliable extraction of static road features. In scenarios with heavy traffic or severe dynamic interferences such as temporary roadblocks, the accuracy and stability of road feature extraction may decline significantly, thereby affecting matching precision. Furthermore, the coexistence of large areas of low-texture regions and localized strong features, along with dynamic noise in urban environments, exacerbates the mismatch rate in the matched point sets. Oh et al. [6] proposed a terrain reference navigation (TRN) method based on LiDAR break-line matching. This study leveraged terrain and intensity information from LiDAR data, along with image matching using the relative edge cross-correlation (RECC) technique, to achieve high-precision navigation for unmanned aerial vehicles (UAVs) in GPS-denied environments (with an average positional accuracy better than 50 cm). However, it should be noted that their method relies on prominent break-line features (such as road edges and building contours), and its applicability and reliability may be constrained in low-texture regions with sparse features. More importantly, the efficiency and robustness of such methods, as well as the fusion algorithm proposed by Xu Haowei et al. [4], in handling low inlier rates during the matching process and subsequent mismatch-removal algorithms, have not been fully validated. Drawing on experience from optical stereo image matching, Shi Xiaotian et al. [5] proposed a complex matching strategy utilizing synthetic aperture radar (SAR) data to achieve stereo SAR matching and high-precision DEM construction, aiming to enhance the performance of terrain-matched navigation. Nevertheless, its computational complexity may pose challenges in meeting the real-time requirements for eVTOL low-altitude high-speed flight applications.

To meet the high-precision navigation and positioning requirements of eVTOLs operating in urban low-altitude environments, this paper focuses on integrating high-precision 3D terrain data from LiDAR and combining the strengths of the SURF and SIFT algorithms. It aims to propose a feature extraction strategy based on enhanced feature description of key regions. In this strategy, the SURF algorithm is employed for global feature extraction to ensure real-time performance, while the SIFT algorithm is utilized in local key regions (areas with abrupt elevation changes) to enhance feature description capabilities. This approach improves matching accuracy while maintaining algorithmic efficiency. Additionally, a self-adaptive RANSAC threshold model based on geospatial resolution is designed. By dynamically adjusting threshold parameters through real-time analysis of terrain relief characteristics, this model overcomes the limitations of traditional fixed thresholds in complex terrains, significantly improving matching robustness and positioning accuracy, in comparison with the adaptive RANSAC algorithm for mismatched-pair detection. This scheme aims to achieve a synergistic improvement in matching accuracy and real-time performance through dynamic feature extraction thresholds and an adaptive misalignment detection mechanism, providing a reliable solution for eVTOL navigation in complex urban low-altitude environments.

## 2. LiDAR-Assisted Terrain-Matching Navigation Framework for eVTOL Aircraft

To achieve high-precision and high-reliability terrain-matched navigation for eVTOL aircraft, especially in highly dynamic and complex urban environments, a LiDAR-assisted terrain-matched navigation framework for eVTOL has been designed, as illustrated in Figure 1. This framework aims to provide eVTOL with precise terrain information and navigation support by constructing a high-precision DEM in real-time and employing optimized terrain feature matching algorithms. The framework comprises two core modules: first, a LiDAR-assisted real-time DEM construction module, and second, a terrain feature matching optimization algorithm module that enhances key region feature description and incorporates adaptive RANSAC mismatch detection.

In complex terrain environments, the navigation accuracy of eVTOL aircraft is highly dependent on precise terrain information. The high-precision DEM constructed in real-time with LiDAR assistance can provide eVTOL aircraft with detailed terrain data, enabling high-precision terrain-matched navigation. Therefore, to construct a real-time elevation map, the following steps are essential:

Firstly, high-precision LiDAR point cloud polar coordinate data must be acquired. In dynamic environments, integrating hardware clock synchronization and sensor calibration compensation, along with synchronously fusing triaxial angular velocity and linear acceleration information output by the inertial measurement unit (IMU), significantly enhances data accuracy and reliability. Subsequently, the extended Kalman filter (EKF) is employed to dynamically estimate the latitude, longitude, altitude, and attitude angles of the carrier, enabling real-time tracking of the aircraft’s state and providing a foundation for subsequent coordinate transformations and DEM construction.

Secondly, the LiDAR point cloud data are transformed from the body frame to the navigation frame (East-North-Up) using the carrier’s attitude matrix, ensuring all data are referenced within a consistent coordinate system, which is crucial for subsequent terrain matching. The sum of the upward component in the East-North-Up coordinate system and the carrier’s altitude is taken as the absolute elevation of the point cloud. The point cloud is then projected into the Earth-Centered Earth-Fixed (ECEF) coordinate system. Utilizing the carrier’s velocity and attitude change rates estimated by the EKF, multiple frames of LiDAR point cloud data are uniformly projected into the navigation frame at the starting moment of the matching window, eliminating point cloud distortions caused by carrier motion and improving DEM construction accuracy.

Finally, the inverse distance weighting (IDW) interpolation method is employed to calculate the elevation value at the center of each grid. GPU acceleration is utilized to enhance real-time performance, and the search radius of IDW is dynamically adjusted to optimize interpolation results based on actual terrain conditions. This provides high-quality data support for subsequent terrain-matched navigation.

Terrain feature matching serves as a pivotal step in the navigation framework. In complex terrain environments, an accurate feature matching algorithm can significantly enhance the precision and reliability of navigation. During the terrain feature matching phase, the real-time DEM constructed in the previous stage is matched with the reference DEM of a specific urban area stored in the eVTOL system. The scale-invariant feature transform (SIFT) and speeded-up robust features (SURF) algorithms are selected to extract local feature points with scale and rotation invariance from both the real-time and reference DEM. To address the shortcomings of the SIFT algorithm’s poor real-time performance and the SURF algorithm’s relatively weak feature description capability, and to enhance the precision and efficiency of terrain feature extraction, a region-adaptive feature selection strategy is adopted. Firstly, the SURF algorithm is employed to rapidly extract terrain features from the elevation map, meeting the demands of real-time navigation. Subsequently, in key frames or key regions (where local terrain curvature or elevation changes exceed predefined thresholds), the SIFT algorithm is utilized to compensate for the SURF algorithm’s inadequacy in feature description within specific areas. This strategy dynamically adjusts the feature extraction method under complex terrain conditions to optimize terrain matching results.

After feature extraction, coarse matching is performed, followed by RANSAC mismatch detection to further refine the matching results and eliminate erroneous matching point pairs. This process enhances both the accuracy and computational speed of terrain-matched navigation.

## 3. Real-Time DEM Construction Based on LiDAR Assistance

The algorithm flow for constructing a real-time DEM assisted by LiDAR is illustrated in Figure 2. Its core steps primarily include sensor data synchronization and preprocessing, dynamic calculation of the carrier’s pose, point cloud coordinate transformation and motion distortion correction, point cloud filtering and fusion, and real-time DEM generation.

### 3.1. Point Cloud Coordinate System Transformation

Point cloud coordinate system transformation is one of the pivotal steps in real-time DEM construction. Its primary objective is to convert the point cloud data acquired by LiDAR from the polar coordinate system to the navigation coordinate system (denoted as the n-frame), facilitating subsequent processing and analysis. This transformation process not only ensures that all data are referenced within a consistent coordinate system but also provides accurate elevation information for terrain matching. Additionally, by eliminating distortions caused by the motion of the carrier, we can further enhance the precision and reliability of the DEM.

Transform the point cloud data (slant range r, azimuth angle α, elevation angle θ) detected by LiDAR from polar coordinates to Cartesian coordinates in the body coordinate system (*b*-frame):(1)$$\left\{\begin{array}{c} x^{b}=r\cos\alpha \cos\theta \\ y^{b}=r\cos\alpha \sin\theta \\ z^{b}=r\sin\theta \end{array}\right.$$

In the formula, r represents the distance from the LiDAR transmitter to the ground target point; α denotes the azimuth angle, which is the angle measured counterclockwise from the forward direction of the eVTOL in the plane parallel to the *XY*-plane of the carrier coordinate system to the projection direction of the target point; and θ signifies the pitch angle, which is the angle measured upward or downward from the horizontal plane of the carrier to the slant range direction line of the target point. This transformation process converts LiDAR data from its original measurement form into a Cartesian coordinate form that is convenient for subsequent processing, providing a foundation for subsequent coordinate transformations and terrain analysis.

Projecting the point cloud onto the navigation frame (*n*-frame) based on the carrier attitude matrix $C_{b}^{n}$ (constructed using roll angle γ, pitch angle β, and yaw angle φ):(2)$$\left[\begin{array}{c} x^{n}\\ y^{n}\\ z^{n} \end{array}\right]=C_{b}^{n}\left[\begin{array}{c} x^{b}\\ y^{b}\\ z^{b} \end{array}\right]$$

Here, $z^{n}+h_{ref}$ represents the absolute elevation of the point cloud, $h_{ref}$ is the geodetic height of the navigation frame origin (initially calibrated through GNSS/barometer), and $z^{n}$ is the upward (altitude) coordinate of the point cloud in the navigation frame (relative height). This transformation process is designed to ensure that all point cloud data are aligned within a unified navigation coordinate system, thereby providing a consistent reference framework for subsequent terrain matching and elevation calculations. Based on the parameters of the WGS-84 Earth ellipsoid model (such as the semi-major axis, flattening, and first eccentricity), calculate the radius of curvature in the meridian ($R_{M}$) and the radius of curvature in the prime vertical ($R_{N}$):(3)$$R_{M}=\frac{a(1-e^{2})}{{\left(1-e^{2}\sin^{2}\varphi \right)}^{3/2}}$$(4)$$R_{N}=\frac{a}{\sqrt{1-e^{2}\sin^{2}\varphi }}$$

Here, the parameters of the Earth’s ellipsoid model include the semi-major axis a, the first eccentricity e, and the geodetic latitude φ.

The radius of curvature in the meridian $R_{M}$ converts a northward displacement increment ($\Delta R_{N}$) into a corresponding latitude variation (Δφ) as follows:(5)$$\Delta \varphi =\frac{\Delta R_{N}}{R_{M}+h}$$
where h represents the geodetic height of the carrier.

The radius of curvature in the prime vertical $R_{N}$ converts the eastward displacement increment ($\Delta R_{E}$) into a latitude variation ($\Delta R_{E}$) as follows:(6)$$\Delta \lambda =\frac{\Delta R_{E}}{\left(R_{N}+h\right)\cos\varphi }$$

Based on the velocity increment $\Delta v_{b}$ in the airborne coordinate system (b-frame) provided by the airborne IMU, combined with attitude information, the velocity increment is transformed into the navigation coordinate system (n-frame) to obtain the velocity increment $\Delta v_{n}$ in the n-frame. Then, by integrating this velocity increment, the displacement increment $\Delta X_{n}$ of the aircraft in the n-frame is calculated:(7)$$\Delta v_{n}=C_{b}^{n}\cdot \Delta v_{b}$$(8)$$\Delta X_{n}=v_{n}\times dt$$

Transform the displacement increment into the navigation frame using the direction cosine matrix C.(9)$$\left[\begin{array}{c} \Delta X_{E}\\ \Delta X_{N}\\ \Delta X_{U} \end{array}\right]=C_{b}^{n}\left[\begin{array}{c} \Delta L_{x}^{b}\\ \Delta L_{y}^{b}\\ \Delta L_{z}^{b} \end{array}\right]$$

Update the carrier’s latitude and longitude:(10)$$\lambda =\lambda_{0}+\Delta \lambda$$(11)$$\varphi =\varphi_{0}+\Delta \varphi$$

The aforementioned steps ensure real-time updates of the carrier’s pose, providing a reference for the dynamic processing of point cloud data. Project multiple frames of point clouds uniformly onto the navigation frame ($n_{0}$ frame) initial moment to eliminate carrier motion distortion:(12)$$\left[\begin{array}{c} x^{n_{0}}\\ y^{n_{0}}\\ z^{n_{0}} \end{array}\right]={\left(C_{n}^{n_{0}}\right)}^{T}\left(\left[\begin{array}{c} x^{b}\\ y^{b}\\ z^{b} \end{array}\right]-P_{n}^{n_{0}}\right)$$

Here, $P_{n}^{n_{0}}$ is the position vector of the carrier in the Earth-Centered, Earth-Fixed (ECEF) coordinate system at the initial moment. By compensating for the carrier’s motion during data acquisition, the impact of motion distortion on the accuracy of DEM is significantly reduced, ensuring high precision and reliability of the real-time elevation map.

### 3.2. LiDAR Point Cloud Filtering and DEM Generation

Within the matching window region defined in the $n_{0}$ frame, the filtered static terrain point cloud is gridded by dividing the point cloud data into equally spaced grid cells based on their coordinates. The elevation of the grid cell centers is then calculated through interpolation using the elevation values of surrounding point clouds. The choice of interpolation method directly impacts the accuracy and practicality of terrain representation. Common interpolation methods include resampling interpolation [26], nearest neighbor interpolation, spline interpolation, and inverse distance weighting (IDW) interpolation. Resampling interpolation achieves efficient computation through grid mean aggregation but struggles to preserve subtle terrain features. The nearest neighbor interpolation method maximally retains the characteristics of the original data but results in a significantly rough elevation surface with poor spatial autocorrelation. Among higher-order interpolation methods, cubic convolution and spline interpolation generate smooth surfaces through global optimization strategies but are sensitive to abrupt terrain changes and prone to over-smoothing or oscillatory distortions. In contrast, the IDW interpolation method balances local feature preservation and computational efficiency through distance-decay weights. Its parameters, such as the power exponent and search radius, can be dynamically optimized based on terrain complexity. This method not only preserves key terrain elements but also ensures algorithm robustness, making it particularly suitable for high-fidelity terrain modeling required in eVTOL low-altitude flight scenarios.

To further enhance the adaptability of IDW interpolation in areas with significant terrain relief and uneven point cloud density, a dynamic adaptive IDW interpolation mechanism is proposed. This mechanism breaks free from the constraints of fixed parameters in traditional IDW and achieves dynamic optimization of interpolation parameters based on local terrain characteristics.

The elevation value is estimated using the IDW interpolation method by calculating the distance weights between the grid point and surrounding points. The elevation estimation formula is as follows:(13)$${\hat{Z}}_{0}(x,y)=\frac{\sum_{i=1}^{N}Z_{i}\cdot d_{i}^{-p(x)}}{\sum_{i=1}^{N}d_{i}^{-p(x)}},\;d_{i}=\sqrt{{\left(x-x_{i}\right)}^{2}+{\left(y-y_{i}\right)}^{2}}$$

Here, $Z_{0}$ represents the estimated elevation of the grid cell center; $Z_{i}$ is the elevation of the *i*-th control point; and $d_{i}$ is the Euclidean distance from the grid cell to the *i*-th control point. p(x) represents an adaptive power exponent that varies with position, and it is calculated based on the elevation variance within the local window surrounding the point:(14)$$p(x)=p_{base}+\alpha \cdot \sigma_{h}^{2}(x)$$

Here, $p_{base}$ is the base power exponent, and α is the adjustment coefficient. When the terrain complexity is high, *p* increases, enhancing the weight of nearby points to preserve edges and details; when the terrain is flat, *p* decreases, improving interpolation smoothness and suppressing noise.

Meanwhile, the search radius *R* is also adaptively adjusted based on the characteristics of local point cloud density:(15)R(x)=k⋅σ(x)

In the formula, $\sigma (x)=\sqrt{\frac{1}{M}\sum_{i=1}^{M}{\left(d_{i}-\bar{d}\right)}^{2}}$ represents the standard deviation of point cloud density within the current window, and *k* is the scaling factor. The search radius *R* is automatically reduced in areas with high point cloud density to preserve more structural details, while it is enlarged in sparse point cloud regions to avoid interpolation voids.

The simulation experiments selected three typical interpolation methods for real-time DEM construction, with results shown in Figure 3. Specifically, Figure 3a illustrates the DEM generated by the nearest neighbor interpolation method, which exhibits distinct blocky features and discontinuities in terrain reconstruction. Figure 3b presents the reconstruction result using the cubic convolution interpolation method, showing relatively smooth terrain overall but with noticeable blurring around building edges. Figure 3c demonstrates the reconstruction effect of the adaptive IDW interpolation method, revealing better preservation of terrain feature edge details, clearer terrain representation, and richer structural information.

Additionally, quantitative performance comparisons of these interpolation methods were conducted, with results shown in Figure 4. Figure 4a compares the root mean square error (RMSE) of DEMs constructed using different methods. It can be observed that the cubic convolution method has the highest RMSE, followed by the traditional IDW and nearest neighbor methods, while the adaptive IDW method achieves the lowest RMSE among all approaches, indicating its significant advantage in elevation reconstruction accuracy. Figure 4b compares edge fidelity performance, where higher values indicate better performance in reconstructing terrain edges. The results show that both traditional and adaptive IDW methods exhibit the best edge fidelity, followed by the nearest neighbor method, with the cubic convolution method performing the weakest.

In summary, the adaptive IDW interpolation method outperforms traditional methods in both DEM reconstruction accuracy and edge preservation capability, effectively supporting real-time DEM construction requirements.

## 4. Research on Terrain-Matching Optimization Algorithm Based on Feature Enhancement of Key Areas

Due to the poor real-time performance of the traditional SIFT feature extraction algorithm [27] and the relatively weak feature description capability of the SURF algorithm [28], in eVTOL applications with high real-time requirements, this study proposes to prioritize the use of the SURF algorithm for feature extraction and coarse matching. Meanwhile, it combines the SIFT algorithm for fine matching in areas with abrupt curvature changes in elevation terrain. This approach aims to enhance the accuracy and stability of matching while ensuring real-time performance, thereby improving both the accuracy and computational speed of terrain matching.

### 4.1. Terrain-Matching Strategy Based on Enhanced Feature Description in Key Areas

To improve the robustness and adaptability of terrain-matching algorithms, particularly in complex urban terrains or areas with weak textures, a terrain-matching strategy based on enhanced feature description in key areas is proposed.

Firstly, perform SURF-based fast global matching and initial pose estimation. The computationally efficient SURF algorithm is employed to rapidly extract features from the real-time DEM constructed in the previous stage. The extracted SURF feature points and their descriptors $d_{SURF}$ are then utilized for rapid feature matching with a pre-stored high-precision reference terrain database. A robust geometric verification based on the RANSAC algorithm is performed on the preliminary matching results to obtain an initial pose estimation and reliable inlier matching pairs $M_{SURF}$.

Secondly, key region identification and SIFT-based fine matching. Utilizing the obtained initial matching results and pose estimation, the real-time sensor image is projected into the coordinate system of the reference elevation map, yielding a projected overlapping region $R_{overlap}$. Within this region, the elevation gradient magnitude of the reference elevation map is calculated.(16)$$G\left(i,j\right)=\sqrt{{\left(\frac{\partial D_{ref}}{\partial x}\right)}^{2}+{\left(\frac{\partial D_{ref}}{\partial y}\right)}^{2}}$$

Based on the gradient map G, a dynamic threshold $T_{gradient}$ is set, and according to this threshold, regions $R_{key}=\left\{\left(i,j\right)|G\left(i,j\right)>T_{gradient}\right\}$ with abrupt elevation changes are identified. The local regions in the projected real-time image that correspond to these identified areas $R_{key}$ are defined as the current key regions $P_{key}$.

Within the identified key areas $P_{key}$, the SIFT algorithm with dynamic thresholds is employed for feature extraction, avoiding full-image SIFT computation and significantly reducing computational time overhead. The SIFT descriptors $d_{SIFT}$ extracted from the key areas are precisely matched with the reference DEM within $R_{key}$, and the SIFT matching results undergo a secondary RANSAC verification to obtain high-precision inlier matching pairs $M_{SIFT}$.

Finally, perform matching information fusion and motion pose estimation. The global SURF inlier matching set $M_{SURF}$ is merged with the SIFT inlier matching set $M_{SIFT}$ from the abrupt elevation change areas to form the final enhanced matching point set:(17)$$M_{enhanced}=M_{SURF}\cup M_{SIFT}$$

Using the fused matching point set $M_{enhanced}$, a robust Levenberg–Marquardt algorithm is employed to perform the final motion pose estimation:(18)$$T=arg\underset{T}{min}\sum_{pi\in P_{final}}\left\|\pi \left(T\cdot P_{i}^{real}\right)-\right.{\left.P_{i}^{ref}\right\|}^{2}$$

Here, $\pi \left(\cdot \right)$ represents the projection function, $P_{i}^{real}$ denotes the real-time 3D point, and $P_{i}^{ref}$ is the corresponding 2D matching point on the reference map.

### 4.2. Adaptive RANSAC Algorithm for Mismatch Detection

To address the threshold sensitivity drawback of the traditional RANSAC algorithm [29] in complex terrain matching, this study proposes a dynamic threshold adjustment method based on geospatial resolution. Traditional methods employ a fixed pixel distance as the threshold for determining inliers, which struggles to adapt to variations in different resolutions and terrain undulations. The maximum inlier distance threshold $d_{max}$ is defined as follows:(19)$$d_{max}=\lambda \frac{\delta_{terrain}}{\delta_{pixel}}$$

Here, $\delta_{terrain}$ represents the standard deviation of terrain elevation for the currently processed local terrain (in meters), $\delta_{pixel}$ denotes the pixel geospatial resolution (in meters per pixel), characterizing the accuracy level of the image, and λ is an adjustment coefficient. The value of the adjustment coefficient λ determines the strictness or leniency of inlier determination. The experiment employs a variety of eVTOL operational scenarios, including urban, hilly, and flat regions, for parameter analysis. As shown in the figure, using the final pose estimation error and inlier ratio as evaluation metrics, the performance of the adjustment coefficient within the range of 0.5 to 3.0 is assessed. The results indicate that when λ<1.2, the threshold becomes overly strict, leading to the erroneous exclusion of a large number of correctly matched points. Consequently, the low inlier ratio results in high pose estimation errors. Conversely, when λ>1.5, the threshold becomes too lenient. Although this increases the inlier ratio, it also significantly raises the false matching rate, again leading to increased pose estimation errors. When $\lambda \in \left[1.2,1.5\right]$, the algorithm maintains both a high inlier ratio and stable pose estimation accuracy. Therefore, the algorithm ultimately selects this intermediate range as the optimal value range for the adjustment coefficient λ and sets λ=1.3 in subsequent experiments.

This model dynamically adjusts the threshold parameter by real-time analysis of terrain relief characteristics, enabling the algorithm to adapt to different terrain complexity scenarios.

To reduce computational overhead, a pre-screening and dynamic iteration module is introduced. Before sampling, a preliminary screening of mismatched points $\left|log\left(scale_{A}/scale_{B}\right)\right|\;>\;0.2$ is conducted based on feature point scale consistency to eliminate obvious mismatched point pairs $M_{gross}$. Subsequently, an early termination condition is set based on the inlier ratio. During the iteration process, the current inlier ratio ω is estimated in real time. If the inlier ratio exceeds a predefined threshold (denoted as ω>0.8), the model is deemed reliable, and the number of iterations is calculated as follows:(20)$$N=\frac{log\left(1-p\right)}{log\left(1-\omega^{n}\right)}$$

Here, *p* represents the desired confidence level, and *n* is the minimum number of samples required to compute the model. If the current number of iterations reaches or exceeds N, the algorithm terminates early.

Finally, by dynamically adjusting threshold parameters through real-time analysis of terrain relief characteristics, the algorithm can adapt to different terrains, thereby improving the matching accuracy in complex scenarios:(21)$$Acurracy=\frac{TP}{TP+FP}\times 100\%$$

Here, TP (true positive) refers to the number of point pairs that are classified as inliers by RANSAC and are indeed correct matches; FP (false positive) refers to the number of point pairs that are classified as inliers by RANSAC but are actually incorrect matches.

### 4.3. Optimized Algorithm Flow for Terrain Matching

Based on the above research content, the specific steps for the study of a terrain-matching optimization algorithm that incorporates a feature extraction strategy enhanced by key region feature description and an adaptive RANSAC-based mismatch detection method are as follows:

Step 1: Utilize LiDAR point cloud data to construct a real-time DEM and load the pre-stored reference DEM from the airborne system.

Step 2: Define a matching window based on the INS position error ellipse to narrow down the search range. Calculate the terrain gradient and curvature in the target area to mark key regions (areas with abrupt elevation changes).

Step 3: First, employ the SURF algorithm to rapidly extract global features. Then, enhance the feature description capability in regions with abrupt elevation changes using the SIFT algorithm.

Step 4: Screen SIFT/SURF feature point pairs based on the nearest neighbor ratio method to narrow down the candidate matching range.

Step 5: Determine if there are sufficient matching points, and then iteratively eliminate mismatching points using the RANSAC algorithm.

Step 6: Integrate laser velocity measurement data to correct the inertial navigation system, output high-precision navigation and positioning results, and continuously provide feedback for correction.

The research flowchart for the terrain-matching optimization algorithm, which incorporates key region feature enhancement and RANSAC-based mismatch detection, is illustrated in Figure 5.

### 4.4. Computational Complexity Analysis of Algorithm

Considering the real-time requirements of eVTOL operational scenarios, this section analyzes the time complexity of the optimized algorithm and compares it with traditional methods. The overall computational complexity of the algorithm is primarily determined by two components: real-time digital elevation model (DEM) construction and feature extraction and matching.
Complexity of DEM construction algorithm.


Let $N_{p}$ denote the number of points in a single-frame LiDAR point cloud, and $N_{g}$ represent the number of target grid cells for interpolation. In the adaptive inverse distance weighting (IDW) interpolation method employed by this algorithm, the computation for each grid cell requires searching for its *k* nearest neighboring points. Consequently, the time complexity of the DEM construction phase is $O(N_{g}\cdot k)$. Notably, the grid interpolation process exhibits high parallelism, through GPU acceleration techniques—effectively achieving near-linear scaling with respect to the search neighborhood size and thereby meeting real-time performance requirements.
2.Complexity of feature extraction and matching algorithm.

Global fast matching stage: Let the number of pixels in the real-time image and the reference image be denoted as $M\cdot log\left(M\right)$ and $N\cdot log\left(N\right)$, respectively. The time complexity of SURF feature extraction is approximately $O(M\cdot log\left(M\right)+N\cdot log\left(N\right))$, while the time complexity of feature description and coarse matching is approximately $O(|F_{surf}^{real}|\cdot |F_{surf}^{ref}|)$, where |F| represents the number of extracted feature points. The complexity of the initial RANSAC validation is $O(I_{1}\cdot C_{1})$, where $I_{1}$ is the number of iterations and $C_{1}$ is the computational cost of model validation per iteration.

Precise matching stage of key areas: This stage represents a critical step in reducing computational overhead for the proposed algorithm. By analyzing the initial matching results, precise calculations are performed only in local regions with abrupt elevation gradients (denoted by a pixel count of $M_{c}$, where $M_{c}\ll M$). Within these regions, the SIFT algorithm is applied, significantly reducing feature extraction complexity from $O(M\cdot log\left(M\right))$ for the entire image to $O(M_{c})$. Subsequently, the complexities of feature matching and secondary RANSAC validation within these local regions are $O(|F_{sift}^{real}|\cdot |F_{sift}^{ref}|)$ and $O(I_{2}\cdot C_{2})$, respectively. Given that both $M_{c}$ and $\left|F_{sift}\right|$ are substantially smaller than their global counterparts, the computational burden in this stage is effectively controlled.

The traditional approach employs the SIFT algorithm throughout the entire process, with its overall complexity for feature extraction and matching expressed as $O(M\cdot log\left(M\right)+N\cdot log\left(N\right)+|F_{sift}^{real}|\cdot |F_{sift}^{ref}|)$. Among these terms, the feature matching component dominates and is typically considered to be of $O(n^{2})$ order.

In contrast, the proposed algorithm achieves significant complexity reduction by employing computationally efficient SURF for global processing while restricting the computationally expensive SIFT to critical regions only. This optimization brings the overall complexity down to approximately(22)$$O(M\cdot log\left(M\right)+N\cdot log\left(N\right)+|F_{surf}^{real}|\cdot |F_{surf}^{ref}|+M_{c}+|F_{sift}^{real}|\cdot |F_{sift}^{ref}|)$$

Given that $M_{c}\ll M$ and $\left|F_{sift}|\ll |F_{surf}\right|$, the algorithm achieves substantial computational efficiency gains. Theoretically, this design ensures both high accuracy and superior computational performance, providing a foundation for meeting the real-time requirements of eVTOL navigation.

## 5. Simulation Experiment and Analysis

### 5.1. Preprocessing of LIDAR Point Cloud Data

For the purpose of constructing a real-time DEM, the algorithm simulation experiment utilized high-resolution topographic data from OpenTopography and LiDAR point cloud data of urban areas included in the tool dataset, sourced from airborne LiDAR surveys (with an RMSE of 0.026 m). The data cover the Whanganui urban area and its surroundings in New Zealand, with a total survey area of 443.52 km^2^ and a point density of 16.58 points per square meter [30]. The reference DEM was generated from this point cloud data, and after interpolation with a 1-m grid, the actual elevation accuracy was influenced by terrain complexity, with an RMSE of approximately 0.5–2 m in urban areas. A typical 1 km × 1 km urban region (containing buildings, roads, green spaces, and other features) was extracted from the reference data. The selected 1 km × 1 km raw point cloud data for the experiment is shown in Figure 6a, and the results of ground point classification after point cloud processing are illustrated in Figure 6b.

Gaussian white noise is added to the theoretical ranging values of the LiDAR data for simulation purposes. The theoretical velocity values are set as the true velocity of the carrier system, while the theoretical ranging values are obtained through ray tracing based on simulated terrain data. The main parameters are set with reference to the Velodyne VLP-16, as shown in Table 1.

In the simulation experiments, the IMU parameters are processed as follows: IMU data are generated through the strapdown inertial navigation inversion algorithm based on the simulated flight trajectory. Accelerometer measurements are simulated by adding Gaussian white noise to the theoretical values, which are synthesized from the acceleration calculated from the flight trajectory and the gravitational acceleration. The main error parameters are referenced from the typical IMU configuration for eVTOL, as shown in Table 2.

During the terrain feature matching stage, multiple sets of baseline DEMs from different terrains (such as urban, mountainous, forested, and plateau regions) are selected. The proposed method is compared with traditional SIFT and SURF algorithms. Both methods are executed separately in the MATLAB R2024a platform, and the accuracy (RMSE) of the matching localization and the average runtime are calculated.

### 5.2. Simulation Experimental Environment

Hardware Platform: Processor—Intel(R) Core(TM) i5-12600KF @ 3.7 GHz, 32.0 GB RAM, GPU—NVIDIA GeForce RTX 4060.

Operating System: Windows 11.

Simulation Software: MATLAB R2024a.

### 5.3. Accuracy Evaluation of DEM

After constructing the DEM, it is necessary to evaluate its accuracy by calculating the error between the constructed real-time DEM and the actual terrain. The experiment compares the constructed DEM with a high-precision DEM generated by airborne LiDAR (with a resolution of 1 m). The root mean square error (RMSE) and mean absolute error (MAE) are selected as evaluation metrics.(23)$$\mathrm{RMSE}=\sqrt{\frac{1}{n}\sum_{i=1}^{n}(z_{pred,i}-z_{ref,i}{)}^{2}}$$(24)$$\mathrm{MAE}=\frac{1}{n}\sum \left|z_{pred,i}-z_{ref,i}\right|$$

Here, $z_{pred,i}$ represents the *i*-th predicted elevation value, and $z_{ref,i}$ represents the *i*-th reference elevation ground truth value.

Since the reference DEM itself contains errors, the calculated RMSE of the DEM under evaluation actually reflects the combined errors of both DEMs:(25)$$RMSE_{observed}=\sqrt{{RMSE_{ref}}^{2}+{RMSE_{test}}^{2}}$$

The experiment compared DEMs constructed using several different interpolation methods, and the experimental results are shown in Figure 7. Compared to the traditional nearest neighbor interpolation method, the resampling method, and the KSSR method proposed by Xinghua Li [31], the DEM constructed using the IDW interpolation method had the smallest errors, with an RMSE of 7.871 m and an MAE of 4.757 m. Additionally, the IDW interpolation method demonstrated the best reconstruction performance at the edges of urban areas. In summary, for the initial stage of real-time DEM construction for eVTOL applications, the IDW interpolation method is the most suitable for constructing DEMs.

### 5.4. Analysis of Terrain Feature Matching Results

During the terrain feature matching stage, the experiment utilized reference DEMs of different terrain regions (such as urban, mountainous, forested, and plateau areas) to perform feature matching with the constructed real-time DEM. The number of features extracted and the number of matching points obtained during the feature extraction stage were statistically analyzed to visualize the effectiveness of the feature extraction algorithm.

The geometric complexity varies significantly across different terrains, such as urban, mountainous, and forested areas. Urban regions, due to their dense buildings, may contain more edge and corner features, whereas forested areas, with their vegetation cover, may result in sparse features. The visualization of the matching map between reference DEM and real-time DEM under different terrain is shown in Figure 8, Figure 8a shows the urban areas, Figure 8b shows the plateau terrains, Figure 8c shows the forested landscapes and Figure 8d shows the mountainous regions. Green Rectangle represents the current confidence search region indicated by the Inertial Navigation System (INS), while Red Circles denote the positions of successfully matched and validated feature points (inliers) on the reference map. The statistical results of the number of features extracted and the number of matching points for DEMs of different terrains are presented in Table 3. Due to the limitations in the resolution of the reference maps and the presence of noise interference in the real-time maps, there is a difference in the number of features between the two. To ensure that the feature density of the reference maps matches that of the real-time maps, the algorithm parameters (such as dynamic thresholds) were further adjusted based on the number of features. The experimental results show that the average number of matching points in each region exceeds 20. If the number of features is too low (e.g., <10), it may lead to matching failure; conversely, an excessive number of features increases the probability of false matches.

After feature extraction, the RANSAC algorithm is employed for mismatch detection. Finally, the matching accuracy of the algorithm is evaluated by calculating the matching accuracy rate (the ratio of the number of inliers after RANSAC filtering to the total number of coarse matching points). The experimental results are illustrated in Figure 9.

The experimental results are shown in Table 4. The optimized algorithm demonstrates significant improvements in both matching rate and localization accuracy: compared with the standard SIFT and standard SURF algorithms, the matching rate increases by 25.0% and 31.4%, respectively, while the positional error decreases by 16.3% and 12.6%, respectively. Although the reduction in RMSE from 1.9 m to 1.59 m represents a relatively modest absolute improvement, its significance in practical applications remains substantial—higher matching success rates and lower localization deviations notably enhance the system’s robustness and usability in complex terrains.

In terms of computational efficiency, although the proposed method, by incorporating a dual-stage SURF/SIFT feature extraction mechanism, exhibits a longer average processing time (34.60 ms) compared to the single-stage SURF-RANSAC approach (14.45 ms), it is significantly faster than the SIFT-RANSAC method (27.45 ms). Furthermore, through GPU-based parallel acceleration techniques (applied to modules such as IDW interpolation and feature matching), the algorithm’s processing time can be optimized to under 20 ms, fully meeting real-time requirements.

To further verify the practical benefits of the accuracy improvements brought by the optimized algorithm, we conducted in-depth statistical analyses of the localization results across multiple scenarios. Although the absolute reduction in RMSE is 0.31 m, the enhanced robustness it demonstrates in practical applications holds significant importance. We evaluated the localization success rates of each algorithm at a 95% confidence level:(26)$$P=\frac{1}{N}\sum_{k=1}^{N}I({\left\|p_{est,k},p_{gt,k}\right\|}_{2}<\delta )\times 100\%$$

Here, $p_{est,k}$ and $p_{gt,k}$ represent the estimated position and the true coordinate position of the k-th frame, respectively; ${\left\|\cdot \right\|}_{2}$ denotes the Euclidean norm; $I\left(\cdot \right)$ is an indicator function; and δ is the error threshold defined for determining position success.

As shown in Table 4, the statistical results indicate that the success rates of the standard SIFT and SURF methods are 76.5% and 82.3%, respectively, whereas the proposed method in this paper significantly increases the success rate to 96.8% (with relative improvements of 20.3% and 14.5%, respectively). This implies that, in the vast majority of operational scenarios, the proposed method can strictly constrain localization errors within an acceptable range, thereby significantly reducing the risk of system failures caused by localization anomalies and providing solid assurance for high-reliability application scenarios.

In summary, the substantial improvements of the proposed method in terms of matching quality, localization accuracy, and environmental adaptability, combined with its acceptable overall time overhead, contribute significantly to the reliability, safety, and practicality of the entire system, fully justifying the reasonableness of its additional complexity.

Through experimental verification and analysis, the feature extraction strategy based on key region feature description enhancement, combined with the optimized adaptive RANSAC mismatch detection algorithm, has achieved a favorable balance between matching accuracy and computational efficiency, providing reliable technical support for terrain-matching-related applications. The current research is primarily validated in a simulated environment. The next step involves conducting tests on a real eVTOL flight platform to evaluate the performance under actual dynamic conditions. Additionally, in-depth research is needed to assess the impact of extreme weather conditions (such as dense fog and heavy rain) on the quality of LiDAR point clouds and the stability of feature extraction. In the future, we can explore the integration of multi-source sensor information, including visual and millimeter-wave radar data, to further enhance the adaptability and robustness of the system in adverse weather conditions or special terrains.

## 6. Conclusions

To address the high-precision navigation requirements of eVTOL aircraft in complex urban low-altitude environments, particularly in GNSS-denied scenarios, this study proposes and validates an optimized terrain-matching algorithm that integrates real-time LiDAR DEM construction, key region feature enhancement, and adaptive RANSAC mismatch detection. The core conclusions are as follows:A method for real-time construction of DEMs based on LiDAR point cloud data is proposed, incorporating a dynamic adaptive IDW interpolation mechanism. This approach overcomes the limitations of fixed parameter settings in traditional IDW methods, significantly improving the accuracy, terrain edge fidelity, and computational efficiency of real-time DEM.A dynamic regional feature extraction strategy is introduced: the SURF algorithm is prioritized to enhance feature extraction speed, while the SIFT algorithm is employed in areas with abrupt elevation changes to improve feature description capability. This strategy significantly optimizes computational resource allocation and enhances the overall algorithm’s operational efficiency. Additionally, a geospatial resolution-based adaptive RANSAC threshold model is designed to overcome the limitations of traditional fixed thresholds in complex terrains, substantially improving matching robustness and localization accuracy.The simulation experimental results demonstrate that the proposed algorithm achieves a localization accuracy better than 2 m in complex scenarios such as typical urban canyons. Compared to traditional SIFT-RANSAC and SURF-RANSAC methods, the matching rate of the proposed algorithm is significantly improved by 25.0% and 31.4%, respectively, while the localization errors are reduced by 16.3% and 12.6%, respectively. Additionally, at a 95% confidence level, the matching success rates of the standard SIFT and SURF methods are 76.5% and 82.3%, respectively, whereas the proposed method significantly elevates the success rate to 96.8% (with relative improvements of 20.3% and 14.5%, respectively), substantially enhancing the navigation and localization accuracy and robustness of eVTOLs in complex low-altitude environments.

## Figures and Tables

**Figure 1 sensors-25-05672-f001:**
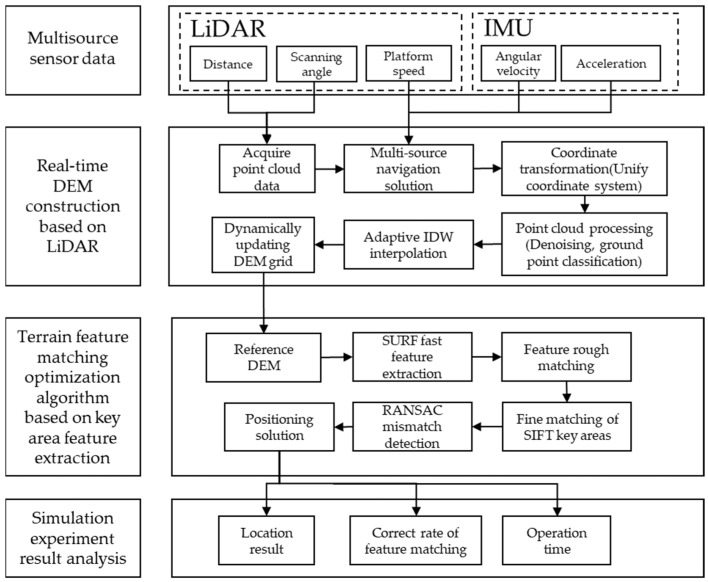
Framework of terrain-matching navigation method for eVTOL based on LiDAR.

**Figure 2 sensors-25-05672-f002:**
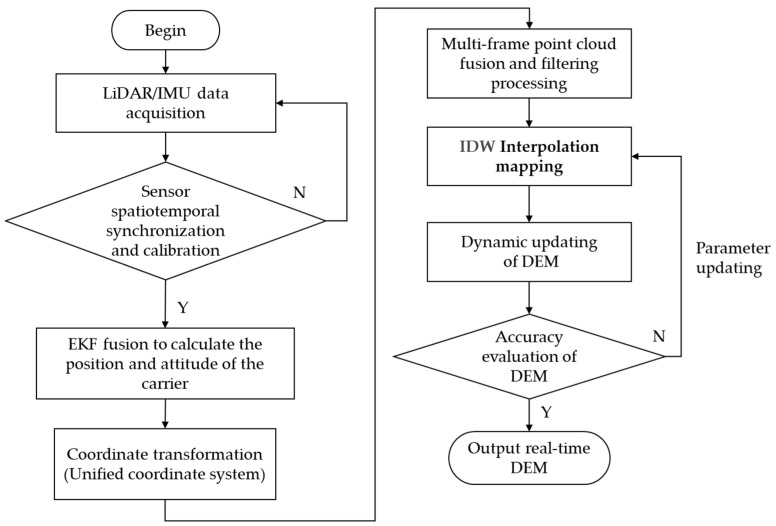
Flowchart of real-time DEM construction algorithm based on LiDAR.

**Figure 3 sensors-25-05672-f003:**
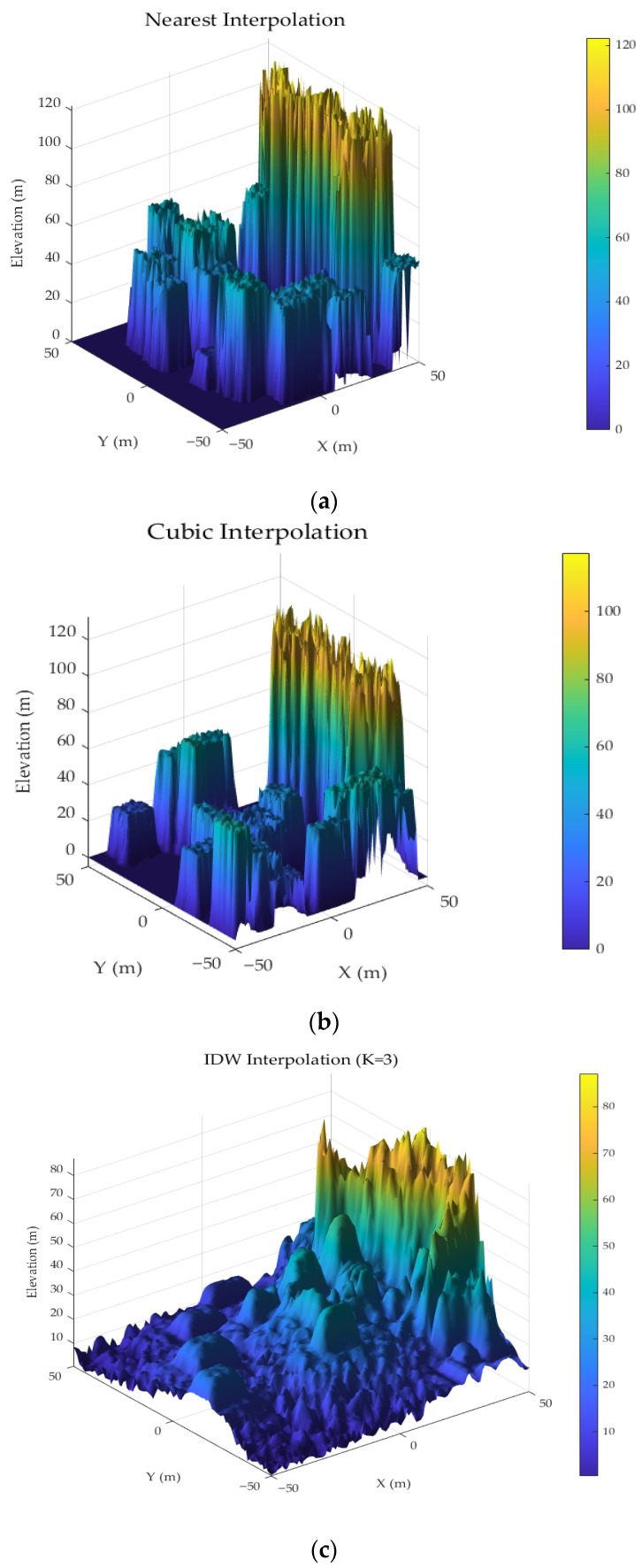
Real-time DEMs constructed using three different interpolation methods. (**a**) nearest Neighbor Interpolation Method; (**b**) cubic convolution Interpolation Method; (**c**) IDW Interpolation Method.

**Figure 4 sensors-25-05672-f004:**
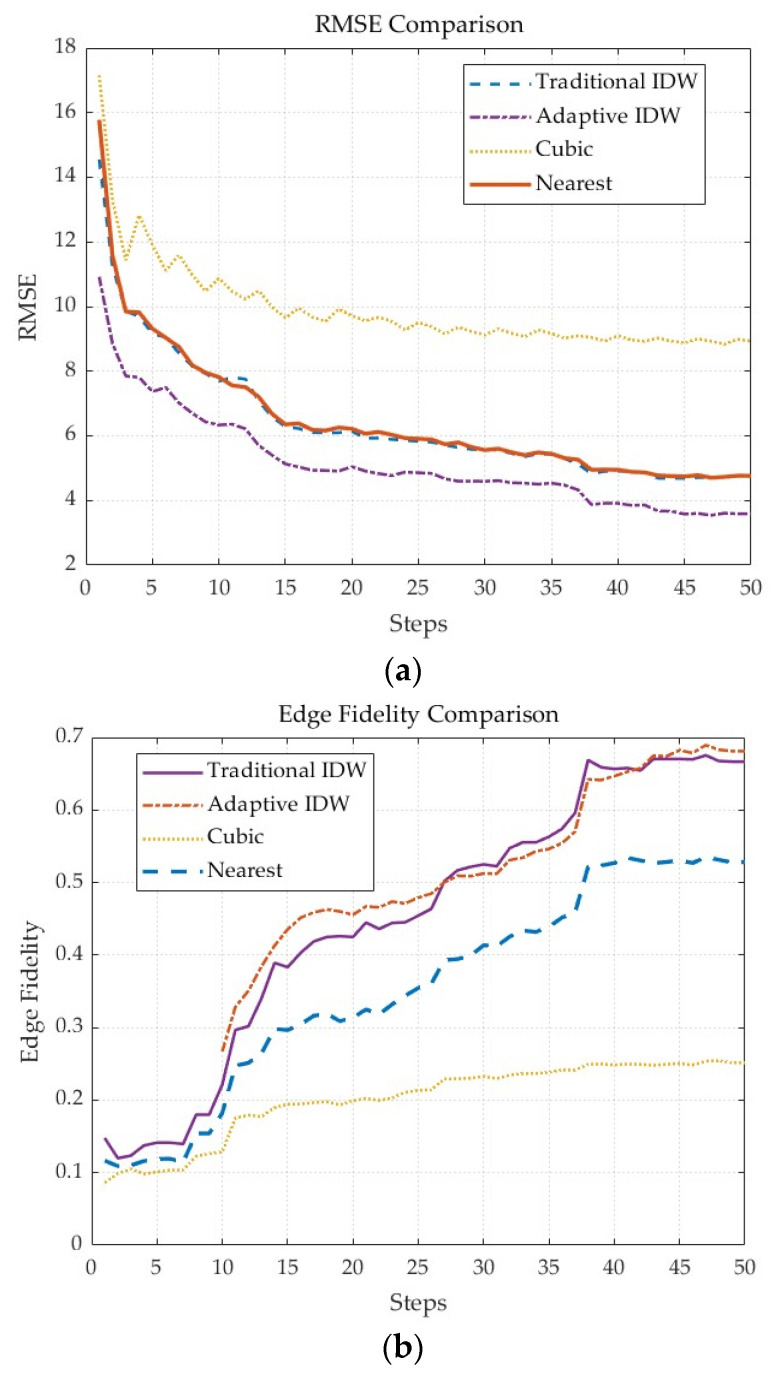
Performance comparison of different interpolation methods. (**a**) RMSE comparison; (**b**) edge fidelity comparison.

**Figure 5 sensors-25-05672-f005:**
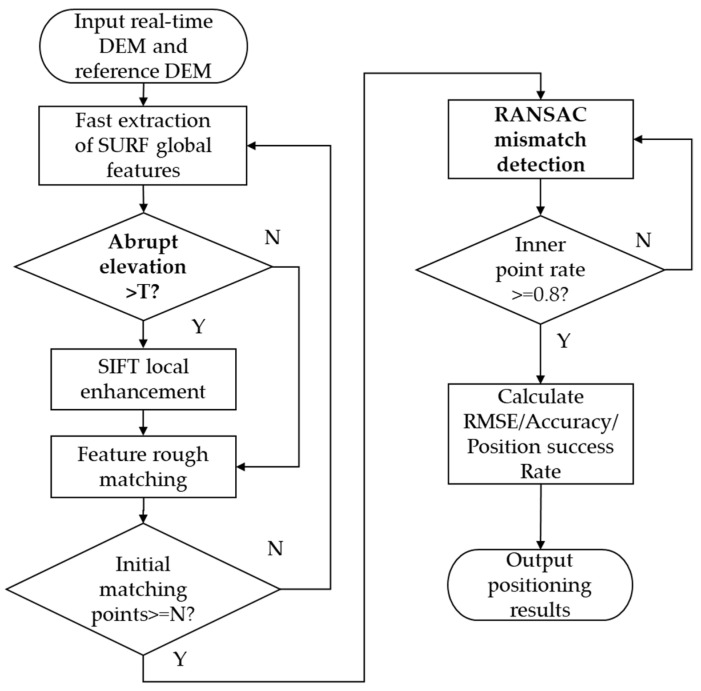
Flowchart of terrain-matching algorithm based on key area feature enhancement and RANSAC mismatch detection.

**Figure 6 sensors-25-05672-f006:**
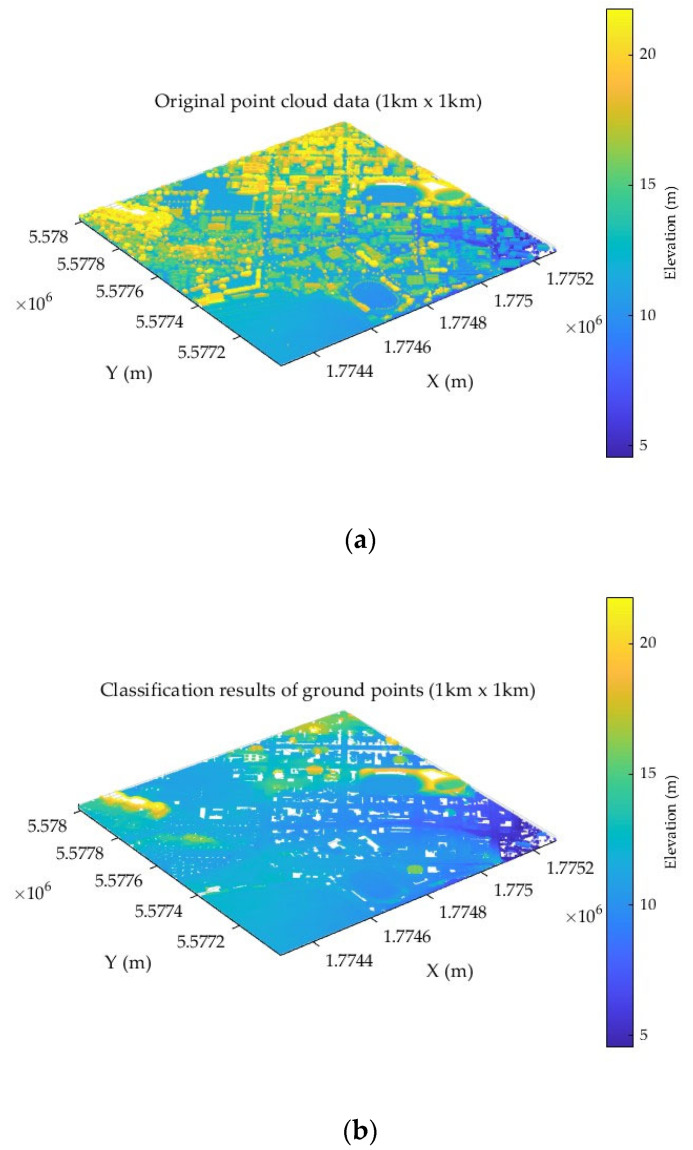
Visualization of point cloud data preprocessing. (**a**) Raw point cloud data; (**b**) results of ground point classification.

**Figure 7 sensors-25-05672-f007:**
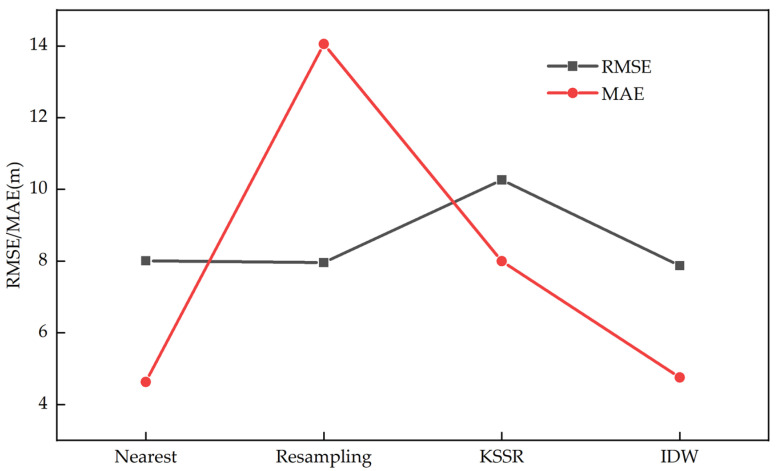
RMSE and MAE corresponding to different interpolation methods.

**Figure 8 sensors-25-05672-f008:**
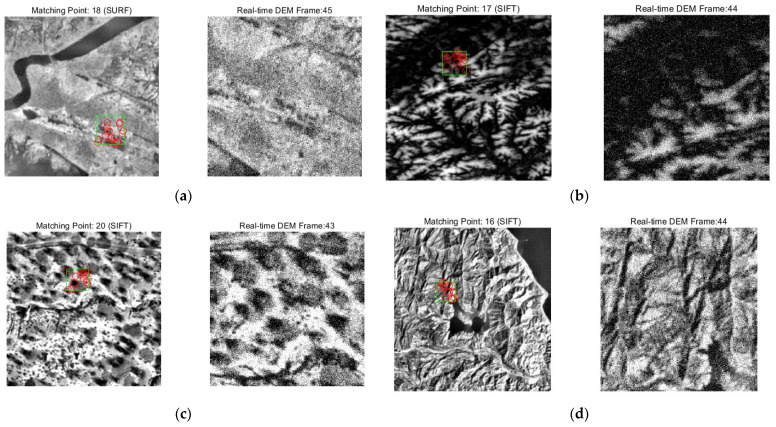
Matching diagram between reference DEM and real-time DEM with different terrain: (**a**) urban areas; (**b**) plateau terrains; (**c**) forested landscapes; (**d**) mountainous regions.

**Figure 9 sensors-25-05672-f009:**
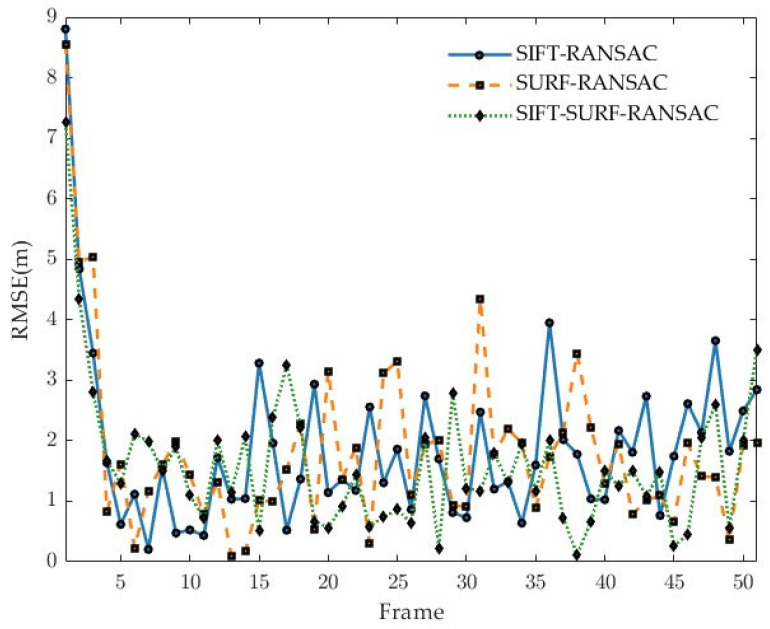
RMSE plot with different methods.

**Table 1 sensors-25-05672-t001:** LiDAR simulation parameters.

Parameter Category	Parameter Values
Frequency	20 Hz
Number of laser beams	16
Horizontal field of view (FOV)	360°
Vertical field of view (FOV)	±15°
Ranging accuracy	±0.026 m

**Table 2 sensors-25-05672-t002:** IMU simulation parameters.

Error Terms	Gyroscope	Accelerometer
Bias	0.1°/h	1 mg
Random noise	0.01°/√h	100 μg/√Hz

**Table 3 sensors-25-05672-t003:** Statistical table of feature counts and matching point counts.

Terrain	Number of Features in Reference DEM	Number of Features in Real-Time DEM	Number of Matching Points
Urban areas	480	510	20
Plateau terrains	353	526	17
Forested landscapes	345	467	23
Mountainous regions	537	599	25

**Table 4 sensors-25-05672-t004:** Performance evaluation of different TAN methods.

Methods	Feature Matching Accuracy (%)	RMSE (m)	Positioning Success Rate (%)	Time Consumption (ms)
SIFT-RANSAC	72.3	1.90	76.5	27.28
SURF-RANSAC	68.8	1.82	82.3	14.45
The method proposed in this paper	90.4	1.59	96.8	34.60

## Data Availability

The LiDAR point cloud data used in this study (accessed on 15th March 2025) are available on the OpenTopography platform (https://opentopography.org) under the dataset DOI: [10.5069/G90Z71GX].

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
