# Peer review of "Research on LiDAR-Assisted Optimization Algorithm for Terrain-Aided Navigation of eVTOL"

_sensors, 2025, doi:10.3390/s25185672_

Round 1
Reviewer 1 Report (Previous Reviewer 1)
Comments and Suggestions for Authors
The authors have revised manuscript based on the reviewer's comments.
Reviewer 2 Report (Previous Reviewer 2)
Comments and Suggestions for Authors
Comments reflected
This manuscript is a resubmission of an earlier submission. The following is a list of the peer review reports and author responses from that submission.
Round 1
Reviewer 1 Report
Comments and Suggestions for Authors
In this paper, the authors proposed a terrain matching optimization algorithm based on LiDAR. Some comments are presented as follows.
- As authors mentioned "The introduction of Light Detection and Ranging (LiDAR) technology provides a breakthrough path to address these issues". The current solution related to LiDAR should be discussed here in terms of pro and con. However, the following paragraphs just summarize the typical works without insights to convince readers.
- Figure 1 gives the framework of terrain matching navigation method for eVTOL based on LiDAR. Do authors follow this framework in the proposed algorithm?
- What is the new in Section 3 as authors list real-time DEM construction?
- The authors do not discuss Figure 3 more in detail.
- Section 4.3 lacks some important equations and discussion.
- What is the computational complexity of the proposed optimization algorithm?
- The main contribution of the proposed optimization algorithm is not clear although the authors presents lots of simulation results.
Author Response
请参阅附件。

Reviewer 2 Report
Comments and Suggestions for Authors
Manuscript offers a detailed breakdown of the DEM construction, feature extraction, and matching strategy. However criticism includes;
-
The explanations are sometimes too procedural (e.g., how point clouds are transformed) without high-level motivation or justification.
-
Several formula derivations are given without sufficient context or assumptions.
-
The proposed algorithm seems more like an incremental integration (SURF + SIFT + adaptive RANSAC) rather than an innovative contribution. It is higly recommended to emphasize why the chosen method (IDW, adaptive thresholds) is optimal beyond standard choices.
-
No quantitative or statistical justification is provided for design choices like the RANSAC λ coefficient range (1.2 to 1.5).
-
The process for identifying key areas could benefit from a more formal definition or algorithmic thresholding approach.
-
All evaluations are based on simulation. No real-world eVTOL data is used.
-
The performance difference (e.g., RMSE reduced from 1.9m to 1.59m) is modest and may not justify the additional complexity unless practical benefits are shown.
Fair
Author Response
请参阅附件。
